# Relationship between Intergenerational Emotional Support and Subjective Well-Being among Elderly Migrants in China: The Mediating Role of Loneliness and Self-Esteem

**DOI:** 10.3390/ijerph192114567

**Published:** 2022-11-06

**Authors:** Man Yang, Hao Wang, Jun Yao

**Affiliations:** 1School of Nursing, Nanjing Medical University, Nanjing 211166, China; 2School of Health Policy and Management, Nanjing Medical University, Nanjing 211166, China; 3Institute of Healthy Jiangsu Development, Nanjing Medical University, Nanjing 211166, China

**Keywords:** subjective well-being, intergenerational emotional support, loneliness, self-esteem, elderly migrant

## Abstract

With the augmentation of family migration, the number and proportion of elderly migrants have increased dramatically in China. The well-being of this group has a profound impact on the whole society. Subjective well-being is a comprehensive reflection of whether a person’s needs are met. In this cross-sectional study, we established a multiple mediation model to evaluate the mediating effects of loneliness and self-esteem on intergenerational emotional support and, consequently, on subjective well-being in elderly migrants. The study population consisted of 728 elderly migrants living in Nanjing (Jiangsu, China), including 219 men (30.1%) and 509 women (69.9%). The participants’ loneliness and self-esteem were measured using the R-UCLA Loneliness Scale, the Rosenberg Self-Esteem Scale, and the Philadelphia Senior Center Confidence Scale. Multiple regression analyses revealed a significant correlation between intergenerational emotional support and subjective well-being, and mediation analysis revealed that intergenerational emotional support indirectly influenced subjective well-being through three mediators: loneliness (mediating effect, 0.149), self-esteem (mediating effect, 0.136), and loneliness and self-esteem (mediating effect, 0.041). We conclude that loneliness and self-esteem mediate the relationship between intergenerational emotional support and subjective well-being in elderly migrants and can be regulated to improve elderly migrants’ subjective well-being. Great attention should be paid to the emotional needs of elderly migrants, and communication and exchange with elderly migrants should be emphasized to enhance their subjective sense of well-being.

## 1. Introduction

With the acceleration of ageing and urbanization in China, an increasing population of elderly migrants, moving mainly from rural areas to cities to live with their children, has emerged [1]. Chinese elderly migrants (aged over 60 years) numbered 13.04 million in 2015, accounting for 5.3% of the total migrant population, and 18 million in 2018, corresponding to 7.5% of the total migrant population. In addition, these numbers keep growing in China [2,3]. Previous studies have explored the barriers that elderly migrants face in applying for public health services in their new cities [4,5,6]. However, few have focused on the emotional and mental health of elderly migrants. Compared to the general elderly, elderly migrants may present social inadaptation, a sense of insecurity, and even symptoms such as depression and anxiety [3,7,8]. These problems may degrade their quality of life and subjective well-being. Therefore, factors that enhance or undermine the subjective well-being of elderly migrants should be explored, in order to establish interventional strategies to protect their mental health. Therefore, we aimed to evaluate the relationships between the following variables in this study.

### 1.1. Intergenerational Emotional Support and Subjective Well-Being

Older adults’ well-being is an important indicator of mental health and quality of life [9], and subjective well-being reflects an individual’s satisfaction with his or her life, including the emergence of positive emotions and the disappearance of negative emotions [9,10]. Factors affecting the subjective well-being are either subjective or objective. Subjective factors involve education level, age, physical status, income, etc. Objective factors involve social support, self-efficacy, self-realization, and self-control [11]. Social support may come from family members, friends, and so on. Intergenerational support has shown a strong relationship with subjective well-being in several studies [12,13]. Intergenerational support is divided into financial support, emotional support, and life care [14]. Intergenerational emotional support refers to daily psychological efforts children take to care for their parents, such as spiritual communication [15]. Intergenerational financial support has also an impact on elderly migrants’ mental health [16,17]. In fact, elderly migrants may need emotional support much more than financial support from their children. Merz suggests that emotional support is generally associated with a higher well-being, while instrumental support with a lower well-being [18]. Several studies have also proposed that non-financial support from children may be more effective than instrumental support to improve the subjective well-being among Chinese elderly [19]. Few studies have been conducted to evaluate the effect of intergenerational emotional support on the subjective well-being of elderly migrants, especially the mediating variables.

### 1.2. Intergenerational Emotional Support, Loneliness and Subjective Well-Being

Loneliness and isolation are common problems among elderly migrants. Compared to younger migrants, elderly migrants are less able to socially integrate and interact. Several studies have shown that elderly migrants have a higher level of loneliness [20]. Loneliness is associated with negative physical and mental health outcomes, especially in older adults [21]. Loneliness also brings a sense of despair and distress [22]. Studies have shown that loneliness has a strong cross-sectional and longitudinal association with subjective well-being [23]. A higher level of loneliness is associated with a lower level of subjective well-being [24]. In contrast, social support, especially that from family members, can improve the subjective well-being and psychological well-being of elderly people [25]. Therefore, we hypothesized that loneliness may be a mediator between intergenerational emotional support and subjective well-being.

### 1.3. Intergenerational Emotional Support, Self-Esteem and Subjective Well-Being

Self-esteem, also termed self-worth, determines one’s perception of quality of life [26]. Self-esteem can be enhanced by emotional support [27]. Emotional support can make one’s value be more recognized by others [28]. Research has shown that self-esteem has a strong positive impact on and can moderately predict subjective well-being [28,29]. Studies based on Europeans and Americans have shown that self-esteem directly and indirectly affects well-being through environmental factors [30,31]. All these studies have verified the positive association between self-esteem and subjective well-being [32,33]. Based on this, we suggest that self-esteem may also mediate the relationship between intergenerational emotional support and subjective well-being.

### 1.4. Present Study

In this study, we explored the mediating effects of loneliness and self-esteem in the relationship between intergenerational emotional support and subjective well-being among elderly migrants in China. Therefore, we designed four hypotheses to develop the model (Figure 1). First, intergenerational emotional support has a positive effect on subjective well-being (H1). Second, loneliness mediates the relationship between intergenerational emotional support and subjective well-being (H2). Third, self-esteem mediates the relationship between intergenerational emotional support and subjective well-being (H3). Finally, loneliness and self-esteem co-mediate the relationship between intergenerational emotional support and subjective well-being (H4).

Although previous studies have elucidated the mechanisms by which intergenerational support enhances subjective well-being, there is room for further exploration. First, few of the previous studies have focused on elderly migrants, a group whose number is increasing year by year. Second, few of the previous studies have explored the relationship here studied using intergenerational emotional support as an independent variable and loneliness and self-esteem as mediating variables. The present study is expected to provide an in-depth understanding of the mechanisms by which intergenerational emotional support may improve the subjective well-being of elderly migrants.

## 2. Methods

### 2.1. Participants

The data in this study were provided by the Social Science Foundation Project of the People’s Republic of China “A follow-up study on the mechanism by which intergenerational relationships influence the mental health of elderly migrants”. This project was performed from September 2019 to September 2020 in Nanjing, China. Elderly migrants were recruited from 21 communities in 7 districts of Nanjing (Qinhuai, Qixia, Gulou, Xuanwu, Jianye, Yuhuatai, and Jiangning), with 3 communities in each district. All participants were voluntary, informed of the purpose of the study, and face-to-face interviewed using a structured questionnaire. All interviewers had experience in medical research and received standardized training for the present study. Only the first-phase survey data were used. A total of 728 elderly migrants were included, comprising 219 (30.1%) males and 509 (69.9%) females.

### 2.2. Measurements

#### 2.2.1. Loneliness

The 3-item R-UCLA Loneliness Scale was used to measure loneliness in the elderly migrants [34]. Each item scores from 1 (hardly ever or never) to 3 (almost always) on a scale, and the total score ranges from 3 to 9. A higher score indicates a higher level of loneliness. The Chinese version of the Scale has shown high reliability and validity among the elderly [35]. The Cronbach’s alpha for the present sample was 0.867.

#### 2.2.2. Self-Esteem

Self-esteem was measured by the Rosenberg Self-Esteem Scale (RSES), which consists of 10 self-esteem-related items such as “On the whole, I am satisfied with myself” and “All in all, I am inclined to feel that I am a failure” [36]. Each item is scored from 1 (strongly disagree) to 7 (strongly agree) on a 7-point Likert-type scale. The RSES score is the sum of the items’ scores reversely coded. The RSES was translated into Chinese and showed good reliability and validity [37]. In this study, the Cronbach’s alpha coefficient for the RSES was 0.851.

#### 2.2.3. Intergenerational Emotional Support

Intergenerational emotional support was evaluated according to the emotional closeness between the elderly migrants and their children. The elderly migrants were asked the following three questions: 1. “Do you feel (emotionally) close to this child?” (Not close = 1, somewhat close = 2, very close = 3); 2. “Do you feel that you get along well with this child?” (Not well = 1, okay = 2, very well = 3); 3. “When you want to talk to this child about your problems or difficulties, do you think he will want to listen to you?” (Not willing = 1, sometimes willing = 2, willing = 3). Scores range from 3 to 9, with a higher score indicating a smoother emotional relationship between the elderly migrants and their children. The Cronbach’s alpha for the present sample was 0.734.

#### 2.2.4. Subjective Well-Being

This variable was measured using the Philadelphia Geriatric Center scale [38], developed by Lawton in 1975. Its Chinese version includes nine items about whether the subject (1) has as much energy as before, (2) is less useful, (3) has a lot of interesting things in life, (4) is depressed, (5) feels happy, (6) has nothing to do, (7) is willing to engage with others, (8) likes to be alone and (9) has become irritable in the past week. Each item is scored 1 (never), 2 (sometimes), and 3 (always). The negative items are reversely coded. A higher score indicates a higher subjective well-being. The Cronbach’s alpha of this scale was 0.74, and the Cronbach’s alpha coefficient of this scale in this study was 0.729.

### 2.3. Data Analysis

In this study, we used the Statistical Package for Social Sciences (SPSS) to complete all data analysis and processing. Firstly, we used descriptive, one-way, and multi-factor analyses of the data. Count data were described as number of cases and percentages; measurement data were described as means and standard deviations; secondly, Pearson correlation was used to test for the correlation between continuous variables; in univariate analysis, statistically significant variables were included as covariates in the moderation model. We used PROCESS macro version 3.5 (model 6) provided by Hayes for the analysis, with intergenerational emotional support as the independent variable, loneliness and self-esteem as mediating variables, and subjective well-being as the dependent variable. A *p* value of 0.05 was considered statistically significant. The study set the bootstrap confidence interval (CI) at 95% based on a bootstrap sample of 5000. If zero was not included in the 95% confidence interval, it indicated a significant mediation effect.

### 2.4. Common Biases of the Method 

We used distributed questionnaires, which could lead to be common methodological bias. Therefore, we adopted methods such as using measurement tools with high reliability and effectiveness and conducting reverse scoring on projects. Prior to the analysis, the common method bias was evaluated by the Hamann single factor test on four questionnaires [39]. Principal component analysis was conducted for all items. The first factor explained the common changes of all items related to each variable. These changes originated from the relationship between the common method deviation and the research variables. If there is no bias, the first factor shows a variation of 25.575%, which is less than 40% of the critical value [40]. Our conclusion was that there was no common method bias in the data.

## 3. Results

### 3.1. Demographic Characteristics of the Participants

The basic characteristics of the participants are shown in Table 1. A total of 728 elderly were included, with an average age of 63.63 ± 5.72 years. Of them, 69.9% were females, 85.9% were married, 50.7% had an education level of junior high school or above, and 68.5% carried a rural Hukou (household registration system in China). A univariate analysis was conducted to evaluate the effects of these characteristics on loneliness. Education level, annual income, Hukou, grandparenting, and physical exercise level were significantly associated with loneliness among the elderly migrants.

### 3.2. Correlation Analysis

The correlation matrix for the key variables is presented in Table 2. Intergenerational emotional support was significantly and positively associated with both subjective well-being and self-esteem (subjective well-being: r = 0.383, *p* < 0.01; self-esteem: r = 0.257, *p* < 0.01). Loneliness was negatively associated with both intergenerational emotional support and self-esteem (intergenerational emotional support: r = −0.263, *p* < 0.01; self-esteem: r = −0.296, *p* < 0.01). Subjective well-being was negatively correlated with loneliness and positively correlated with self-esteem (loneliness: r = −0.432, *p* < 0.01; self-esteem: r = 0.496, *p* < 0.01).

### 3.3. Multiple Mediation Model

The multiple mediation model was constructed to explore the mediating effects of loneliness and self-esteem among the elderly migrants. The variables included gender, age, education level, smoking status, and alcohol consumption. Intergenerational emotional support was set as an independent variable, and subjective well-being as a dependent variable. Loneliness and self-esteem were estimated as two mediators.

The analysis (Table 3) showed that intergenerational emotional support was negatively and positively associated with loneliness (a1 = −0.263, *p* < 0.001) and self-esteem (a2 = 0.673, *p* < 0.001), respectively. Loneliness was negatively correlated with self-esteem (d1 = −0.769, *p* < 0.001). In addition, loneliness (b1 = −0.568, *p* < 0.001) and self-esteem (b2 = 0.202, *p* < 0.001) were negatively and positively correlated with subjective well-being, respectively. Intergenerational emotional support was positively correlated with subjective well-being (c’ = −0.512, *p* < 0.001).

The mediation pathways are shown in Figure 2. The pass-through coefficients indicated that all relationships in the model were significantly positively and negatively correlated. The direct effect of intergenerational emotional support on subjective well-being remained significant after the inclusion of loneliness and self-esteem, indicating that both mediators linked intergenerational emotional support with subjective well-being.

## 4. Discussion

This study verified that loneliness and self-esteem mediate the relationship between intergenerational emotional support and subjective well-being among elderly migrants in China. Significant correlations exist between intergenerational emotional support, loneliness, self-esteem, and subjective well-being. These findings provide more details about the relationship between intergenerational support and subjective well-being (see Table 4 for more information).

### 4.1. Role of Intergenerational Emotional Support

As shown by the regression model, intergenerational emotional support significantly and positively affected subjective well-being, with a direct effect of 61.10%. This result is consistent with that from a previous empirical study exploring the effect of intergenerational emotional support on subjective well-being among elderly migrants in China [41]. Numerous studies have shown that a lack of emotional support increases the risk of mental illnesses [42]. Subjective well-being and age are closely related, and respondents from Latin America showed a gradual decline in well-being with age [43]. In China, one-third of older adults have symptoms of depression and anxiety, which greatly reduce their subjective well-being and quality of life [44]. Adequate emotional support should be given to improve the mental health of elderly migrants in China.

### 4.2. Mediation Effect of Loneliness

As shown by the multiple mediation model, loneliness and self-esteem linked intergenerational emotional support with subjective well-being, suggesting that intergenerational emotional support enhances subjective well-being by reducing loneliness as well as by increasing self-esteem. The total mediating effect of the two mediators was 38.90%, indicating that both play a key role in the relationship between intergenerational emotional support and subjective well-being. Moreover, loneliness exerted the strongest mediating effect of 17.78%. Therefore, hypothesis 2 of this study was confirmed: intergenerational emotional support can enhance subjective well-being by reducing loneliness in elderly migrants. Intergenerational emotional support can provide motivation to the elderly to participate in social activities, which in turn relieves their loneliness and enhances their subjective well-being [45,46]. With emotional support from their children, elderly migrants become more confident in communicating with others in new environments [47,48]. Therefore, the children should provide emotional support to the elderly who migrated to new places [49].

### 4.3. Mediation Effect of Self-Esteem

We also found that self-esteem mediated the relationship between intergenerational emotional support and subjective well-being among elderly migrants in China. Therefore, hypothesis 3 was also confirmed. Intergenerational emotional support can enhance subjective well-being by increasing the self-esteem of elderly migrants. The mediating effect of this path was 16.23%, suggesting that self-esteem is strongly related to subjective well-being. This may be explained by the fact that people with a high level of self-esteem are more positive or optimistic in facing difficulties [50,51,52]. The elderly migrants with a higher level of intergenerational emotional support also showed a higher level of self-esteem. As mentioned above, intergenerational emotional support can increase the confidence of elderly migrants in social communication, which benefits the development of a sense of self-esteem [53]. Together, intergenerational emotional support can enhance subjective well-being through increasing the self-esteem of elderly migrants.

### 4.4. Serial Mediation Effect of Loneliness and Self-Esteem

Our study showed that loneliness and self-esteem played serial mediating roles between intergenerational emotional support and subjective well-being, which proved the trueness of hypothesis 4. Intergenerational emotional support can enhance subjective well-being through serially reducing loneliness and increasing self-esteem. Migration can aggravate the loneliness of the elderly, especially when emotional support from their families lacks [48,54]. According to a sociometric theory, people with a feeling of isolation or interpersonal rejection tend to exhibit a lower self-esteem [55]. A higher loneliness brings a lower self-esteem in Cacioppo’s study, and this is associated with a decrease in well-being [56]. A poor intergenerational emotional support can increase loneliness and undermine self-esteem, thus impairing subjective well-being.

### 4.5. Limitations

Some limitations of this study need to be acknowledged. First, the data collected through questionnaires and evaluated by scales in the form of “face-to-face” interviews may be subjective, and multiple methods should be used to collect information. Second, this was a cross-sectional study, so future experimental and longitudinal studies should be conducted to examine the relationship between intergenerational emotional support, loneliness, self-esteem, and subjective well-being. Third, the data were only collected from elderly migrants in Nanjing, and subsequent studies should expand the sample size and study area to obtain more representative data.

## 5. Conclusions

Our study shows that loneliness and self-esteem sequentially mediate the positive relationship between intergenerational emotional support and subjective well-being in Chinese elderly migrants. The subjective well-being of elderly migrants can be enhanced through strategies to increase intergenerational emotional support. First of all, the government should pay attention to the mental health of these migrant elderly, optimize the social environment of the area they have moved to, use community services to build a communication platform for them, and implement activities to promote family communication. Secondly, migrant children should increase their emotional communication with their parents, understand their elders’ inner world, ease their loneliness, and offer them understanding and support. In future studies, larger samples should be examined to verify our findings.

## Figures and Tables

**Figure 1 ijerph-19-14567-f001:**
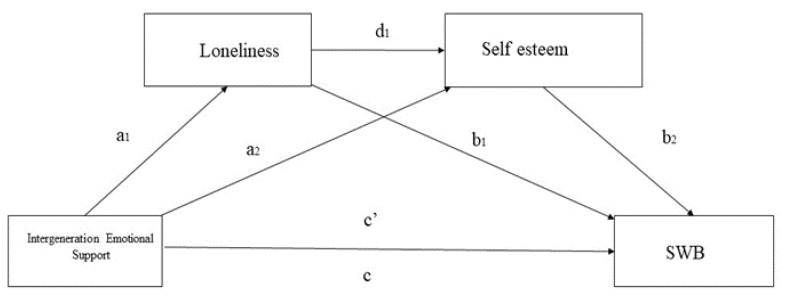
Model to test our hypotheses. Note: a, b, c, c’, and d represent path coefficients. Note1: a1: represents the path from IES to loneliness; a2: represents the path from IES to; b1 represents the path from loneliness to SWB; b2 represents the path from Self-esteem to SWB; d1 represents the path from loneliness to Self-esteem. Note2: SWB represents subjective well-being.

**Figure 2 ijerph-19-14567-f002:**
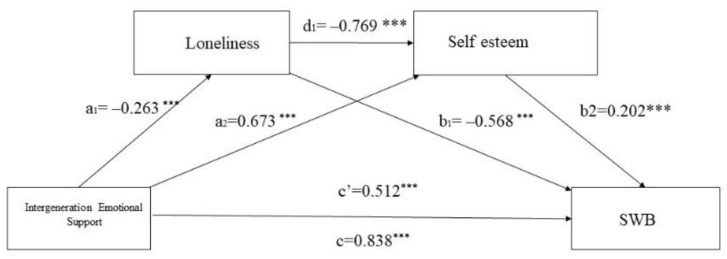
Conceptional framework of the multiple mediation model. Note: *** *p* < 0.001. Note1: a1: represents the path from IES to loneliness; a2: represents the path from IES to; b1 represents the path from loneliness to SWB; b2 represents the path from Self-esteem to SWB; d1 represents the path from loneliness to Self-esteem. Note2: SWB represents subjective well-being.

**Table 1 ijerph-19-14567-t001:** Demographic characteristics of the participants (*n* = 728).

Variables	*n*	%	*p*-Value
Total	728	100	
Gender			0.29
Male	219	30.1	
Female	509	69.9	
Marital status			0.027
Currently married	625	85.9	
Single ^a^	103	14.1	
Educational level			<0.001
Illiterate	156	21.4	
Primary school	203	27.9	
Middle school or above	369	50.7	
Year income ^b^			<0.001
Q1	401	55.1	
Q2	236	32.3	
Q3	91	12.5	
Hukou			<0.001
Rural	499	68.5	
Urban	229	31.5	
Religious faith			0.231
Yes	105	14.4	
No	623	85.6	
Take care of every generation			<0.001
Yes	583	80.1	
No	145	19.9	
Perform physical exercise			<0.001
Yes	462	63.5	
No	266	36.5	

Notes: ^a^ “Single” included those who were divorced and widowed; ^b^ Q1 (Less than 10,000 yuan) Q2 (10,000 to 40,000 yuan), Q3 (Over 40,000 yuan).

**Table 2 ijerph-19-14567-t002:** Descriptive statistics and correlations among the key variables (*n* =728).

Variables	M ± SD	1	2	3	4	5	6	7
1. gender	1.700 ± 0.459	1						
2. marital status	1.160 ± 0.412	0.021	1					
3. education background	2.600 ± 1.186	−0.155 **	−0.093 *	1				
4. intergenerational Emotional support	7.850 ± 1.318	0.079 *	−0.025	0.102 **	1			
5. loneliness	4.030 ± 1.407	−0.020	0.130 **	−0.151 **	−0.263 **	1		
6. self-esteem	37.620 ± 5.156	0.054	−0.095 *	0.267 **	0.257 **	−0.296 **	1	
7. subjective wellbeing	21.820 ± 3.090	0.039	−0.099 **	0.238 **	0.383 **	−0.432 **	0.496 **	1

Note: * *p* < 0.05, ** *p* < 0.01.

**Table 3 ijerph-19-14567-t003:** Regression coefficients in the serial mediation analysis (*n* = 728).

Criterion	Predictors	R	R^2^	F	B	t	95%CI
loneliness	IES	0.313	0.098	19.626	−0.263 ***	−6.905	(−0.338, −0.188)
self-esteem	IES	0.418	0.175	30.524	0.673 ***	4.875	(0.402, 0.944)
	loneliness	-	-	-	−0.769 ***	−5.894	(−1.025, −0.513)
intergenerational	IES	0.621	0.385	75.317	0.511 ***	7.045	(0.369, 0.654)
emotional support	Loneliness	-	-	-	−0.568 ***	−8.308	(−0.703, −0.432)
	Self-esteem	-	-	-	0.202 ***	10.505	(0.165, 0.240)

Note: IES: intergenerational emotional support; *** *p* < 0.001.

**Table 4 ijerph-19-14567-t004:** Indirect effects of activities of intergenerational emotional support on subjective wellbeing, with loneliness and self-esteem as mediators (*n* = 728).

Model Pathways	B	Boot SE	Mediating Effect	95% CI
Total effect IES →subjective wellbeing	0.838 ^a^	0.080	100%	(0.682, 0.993)
Direct effect IES→ subjective wellbeing	0.512 ^a^	0.073	61.10%	(0.369, 0.654)
Total indirect effect IES→ subjective wellbeing	0.326 ^a^	0.046	38.90%	(0.240, 0.416)
IES→ Loneliness→ subjective wellbeing	0.149 ^a^	0.031	17.78%	(0.092, 0.213 )
IES→ self-esteem→ subjective wellbeing	0.136 ^a^	0.031	16.23%	(0.078, 0.198)
IES→ Loneliness→ self-esteem →subjective wellbeing	0.041 ^a^	0.011	4.89%	(0.022, 0.065)

Note: ^a^ Empirical 95% confidence interval does not overlap with zero.

## Data Availability

The datasets used and analysed in this study are available from the corresponding author on reasonable request.

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
