# Peer review of "Relationship between Intergenerational Emotional Support and Subjective Well-Being among Elderly Migrants in China: The Mediating Role of Loneliness and Self-Esteem"

_ijerph, 2022, doi:10.3390/ijerph192114567_

Round 1
Reviewer 1 Report
1. What is the main question addressed by the research?
The main aim of this study is to examine factors that improve or undermine the subjective well-being of elderly migrants in order to develop intervention strategies to protect the mental health of elderly migrants.
The authors proposed four hypotheses:
H1 - intergenerational emotional support has a positive effect on subjective well-being.
H2 - loneliness mediates the relationship between intergenerational emotional support and subjective well-being.
H3 - self-esteem mediates the relationship between intergenerational emotional support and subjective well-being.
H4 - loneliness and self-esteem co-mediate the relationship between intergenerational emotional support and subjective well-being.
2. Do you consider the topic original or relevant in the field? Does it address a specific gap in the field?
The Authors address the issue of internal migration of people over 60 years of age, which is considered not only from the social side, but above all from the psycho-emotional state of these people, which is an important society problem.
3. What does it add to the subject area compared with other published material?
The study population consisted of 728 elderly migrants, which is a significant indicator. Three baseline variables were selected to examine the research problem: intergenerational emotional support, loneliness of elderly migrants and self-esteem (self- worth). The study investigated both the direct influence of the variables and their multiple relations.
4. What specific improvements should the authors consider regarding the methodology? What further controls should be considered?
In order to optimise the structure of the abstract, the relevance of the research topic should have been added at the beginning of the abstract.
The paper would have looked better if the abstract had also indicated the effects of the research carried out - i.e. what intervention strategies should look like to protect the mental health of elderly migrants or what recommendations could be offered to elderly migrants in China to strengthen their well-being.
5. Are the conclusions consistent with the evidence and arguments presented and do they address the main question posed?
The paper is well structured, with a clearly stated purpose and research hypotheses. However, the conclusions should have focused a little more broadly on the Authors' vision of improvements aimed at enhancing intergenerational emotional support.
6. Are the references appropriate?
The references presented in this paper are relevant to the current state of research on the issues discussed. However, the editorial requirements regarding the presentation of the references in the text of the paper have not been met.
7. Please include any assitional comments on the tables and figures.
There are no critical comments on the tables and figures.
Author Response
Dear editorial office:
With this cover letter, I will submit the revised manuscript (IJERPH-2009084) entitled "Relationship between intergenerational emotional support and subjective well-being among the elderly migrants in China: the mediating role of loneliness and self-esteem" I would like to thank referees for the careful and constructive reviews. Based the comments from the referees, I have made changes of the manuscript which are detailed below.
Response to Reviewer 1Comments
Point 4: To optimize the structure of the abstract, the relevance of the research topic should have been added at the beginning of the abstract. The paper would have looked better if the abstract had also indicated the effects of the research carried out. what intervention strategies should look like to protect the mental health of elderly migrants or what. recommendations could be offered to elderly migrants in China to strengthen their well-being.
Response 4:Yes, thank you for your suggestion. Based on your comments, I have added contents related to the research topic in the abstract section, such as the gradually large number and scale of elderly migrants in China, and as a special group of elderly, their psychological problems should be taken seriously by the government and families. By analyzing the data, I conclude that intergenerational emotional support has a profound impact on the subjective well-being of elderly migrants and that migrant children should increase communication with their parents and pay attention to their parents' emotional needs.
Point 5: The paper is well structured, with a clearly stated purpose and research hypotheses. However, the conclusions should have focused a little more broadly on the Authors' vision of improvements aimed at enhancing intergenerational emotional support.
Response 5:Thank you. In view of your suggestion, I believe that the government should first rely on community and other grass-roots organizations to establish groups for elderly migrants, and regularly hold relevant activities. In this way, the opportunities for communication with people are increased and the sense of loneliness is alleviated; Floating children should not only give their parents material support, but also care about their parents' spiritual world, and carry out some activities to promote family harmony to enhance their sense of happiness.
Point 6: The references presented in this paper are relevant to the current state of research on the issues discussed. However, the editorial requirements regarding the presentation of the references in the text of the paper have not been met.
Response 6:Thank you. I noticed that the references do not conform to the requirements of the magazine editor. I have corrected this error in the revised edition.
I deeply appreciate your comments on my manuscript. Thank you for reviewing my manuscript.
Sincerely.
Man Yang
Reviewer 2 Report
This is a very interesting and meaningful study with a large sample size. Judging from the overall length and wording of the article, it is undoubtedly rich and rigorous, especially in the introduction and discussion sections. Readers are well presented with several sets of related concepts involved in this study, and statistical methods are used appropriately. But my concerns are as follows:
Abstract:
I personally recommend a little less content in this section, if possible, please increase this section; if the journal requires a word limit, then please ignore this suggestion.
Methods:
Measurements: "2.3.3 Intergenerational emotional support" This section seems to be missing the display of the Cronbach coefficient, please review it.
Discussion:
This part shows up Table4, which seems to still be showing data from the Results section. Perhaps this is intentional by the author to better follow the Discussion section, but I still recommend that the author move Table4 and its related content to the Results section.
Funding: The author wrote "No" in this section, but in the previous section you mentioned the National Social Science Fund, please review it.
Other Opinions:
1. Some spell errors, blank space, and grammar issues should be checked throughout the manuscript.
2. The format of the references needs to be double-checked to meet the requirements for publication
Author Response
Dear editorial office:
With this cover letter, I will submit the revised manuscript (IJERPH-2009084) entitled "Relationship between intergenerational emotional support and subjective well-being among the elderly migrants in China: the mediating role of loneliness and self-esteem" I would like to thank referees for the careful and constructive reviews. Based the comments from the referees, I have made changes of the manuscript which are detailed below.
Response to Reviewer 2 Comments
Point 1 Abstract: I personally recommend a little less content in this section, if possible, please increase this section; if the journal requires a word limit, then please ignore this suggestion
Response1: Yes, thank you for the suggestion. This suggestion is similar to that of reviewer 1. Based on the this, I added contents related to the research topic in the abstract section, such as the gradually large number and scale of elderly migrants in China, and as a special group of elderly people, their psychological problems should be paid attention to by the government and families. Through the analysis of the data, I conclude that intergenerational emotional support has a profound impact on the subjective well-being of elderly migrants, and children of migrants should strengthen communication with their parents and pay attention to their parents' emotional needs.
Point 2 Methods: Measurements: "2.3.3 Intergenerational emotional support" This section seems to be missing the display of the Cronbach coefficient, please review it.
Response 2: Thanks for the heads up, I've submitted the missing content in the revised version.
Point 3 Discussion: This part shows up Table4, which seems to still be showing data from the Results section. Perhaps this is intentional by the author to better follow the Discussion section, but I still recommend that the author move Table4 and its related content to the Results section.
Response 3: Thank you for the suggestion. I've changed the location of table 4 in the revised version.
Point 4 Funding: The author wrote "No" in this section, but in the previous section you mentioned the National Social Science Fund, please review it.
Response 4: Thank you for the reminder. I have added the fund name of the study in the revised version.
I deeply appreciate your comments on my manuscript. Thank you for reviewing my manuscript.
Sincerely.
Man Yang